# CO$_2$ Electroreduction over Metallic Oxide, Carbon-Based, and Molecular Catalysts: A Mini-Review of the Current Advances

Hassan Ait Ahsaine [1,*] , Mohamed Zbair [2,3] , Amal BaQais [4] and Madjid Arab [5,*]

1   Laboratoire de Chimie Appliquée des Matériaux, Faculty of Sciences, Mohammed V University in Rabat, Rabat 1014, Morocco
2   Mulhouse Materials Science Institute (IS2M), Université de Haute-Alsace, CNRS, UMR 7361, F-68100 Mulhouse, France; mohamed.zbair@uha.fr
3   Département de Chimie, Université de Strasbourg, F-67000 Strasbourg, France
4   Department of Chemistry, College of Science, Princess Nourah Bint Abdulrahman University, P.O. Box 84428, Riyadh 11671, Saudi Arabia; aabaqeis@pnu.edu.sa
5   Aix Marseille Univ, Univ Toulon, CNRS, IM2NP, CS 60584, CEDEX 9, F-83041 Toulon, France
*   Correspondence: h.aitahsaine@um5r.ac.ma (H.A.A.); madjid.arab@univ-tln.fr (M.A.)

**Abstract:** Electrochemical CO$_2$ reduction reaction (CO$_2$RR) is one of the most challenging targets of current energy research. Multi-electron reduction with proton-coupled reactions is more thermodynamically favorable, leading to diverse product distribution. This requires the design of stable electroactive materials having selective product generation and low overpotentials. In this review, we have explored different CO$_2$RR electrocatalysts in the gas phase and H-cell configurations. Five groups of electrocatalysts ranging from metals and metal oxide, single atom, carbon-based, porphyrins, covalent, metal–organic frameworks, and phthalocyanines-based electrocatalysts have been reviewed. Finally, conclusions and prospects have been elaborated.

**Keywords:** electrocatalysis; CO$_2$ reduction; molecular electrocatalysts; CO$_2$ conversion





## 1. Introduction

Carbon dioxide (CO$_2$) is the most well-known greenhouse gas, which is produced both naturally and artificially. It is also required for the growth of all plants on the planet, as well as many industrial operations [1]. In an ideal world, CO$_2$ generated on Earth would be balanced by CO$_2$ consumed, ensuring that CO$_2$ levels remain constant and environmental stability is maintained. Unfortunately, as human industrial activities have become more intense, this equilibrium has been broken, resulting in increased CO$_2$ generation and making global warming an urgent concern. As a result, limiting CO$_2$ production and turning CO$_2$ into usable materials appears to be vital, if not critical, for environmental protection, and numerous governments throughout the world have shown their concern by boosting research funding to address the CO$_2$ problem. Numerous research studies have focused on the development and implementation of renewable energy sources as a way to reduce reliance on fossil fuels [2,3], as well as CO$_2$ capture and utilization technologies. CO$_2$ usage would minimize greenhouse gas emissions in the atmosphere and seas, where they might cause harm, and CO$_2$ could also be utilized to make valuable compounds [4–6].

As CO$_2$ is the most thermodynamically stable carbon molecule, it needs a lot of energy to transform into value-added compounds. Various chemical reactions have been described that can convert CO$_2$ into compounds such as CO, hydrocarbons, or oxygenated hydrocarbons. Gas-phase reactions, liquid-phase reactions, electrochemical reactions, and photocatalytic reactions have all been described. Gas-phase activities include dry reforming of methane (CH$_4$ + CO$_2$ → 2CO + 2H$_2$) and hydrogenation of CO$_2$ (CO$_2$ + H$_2$ → CO + H$_2$O, commonly known as the water-gas shift reversal reaction;

$CO_2 + 4H_2 \rightarrow CH_4 + 2H_2O$). The liquid phase technique uses $CO_2$ dissolved in an aqueous phase ($CO_2$ (aq) + $H_2$ (aq) $\rightarrow$ COOH) to make formic acid. Several review studies on $CO_2$ hydrogenation have been published [5–8]. The hydrogenation of $CO_2$ or the synthesis of formic acid, on either side, necessitates $H_2$, which is most typically generated from methane by steam reforming, which also generates a significant quantity of $CO_2$.

Although electrochemical $CO_2$ reduction has garnered a lot of interest recently, the poor solubility of $CO_2$ in aqueous solutions has been a major impediment. The use of a gas diffusion electrode has made it possible to employ gaseous $CO_2$ directly for electrochemical conversion. $H_2$ is not required as a reactant in this electrochemical conversion process. The electrochemical reduction of $CO_2$ is not only a practical way to utilize $CO_2$, but also a promising future alternative for storing intermittent energy from renewable sources, since it allows electrical energy to be stored in the form of chemical bonds. Carbon monoxide (CO), formic acid (HCOOH), formaldehyde ($CH_2O$), methanol ($CH_3OH$), methane ($CH_4$), ethanol ($C_2H_6O$), ethylene ($C_2H_4$), or n-propanol ($C_3H_8O$) are some of the important byproducts of $CO_2$ electroreduction [1,8]. The hurdles of converting $CO_2$ to $CH_3OH$ are particularly daunting, but the potential benefits are huge, as $CH_3OH$ has a very high energy density and is a crucial intermediary for various bulk chemicals used in everyday items such as silicone, paints, and plastics [9,10].

In addition, the majority of electrodes used in $CO_2$ electroreduction are metal plates, metal granules, or electrodeposited metals on a substrate [1]. The mass transfer of $CO_2$ from the bulk to the solid electrode surface, however, is limited by the comparatively low solubility of $CO_2$ in water under ambient circumstances, thus limiting the reaction rates and current densities of $CO_2$ electroreduction [11,12].

This review highlights the recent progress in the field $CO_2$ electroreduction including the use of gas diffusion electrode configuration (GDE). We have reviewed the different electrode materials ranging from oxides, metallic and bimetallic, carbon-based materials, single-atom catalysts, and molecular catalysts (porphyrins, metal–organic frameworks, covalent organic frameworks, and phthalocyanines) for the $CO_2$ reduction to $C_1$ and $C_2$ chemicals. The effect of several parameters was discussed.

## 2. Electrocatalytic $CO_2$ Reduction

### 2.1. Oxide, Metallic, and Bimetallic Catalysts

The use of gas-diffusion electrodes (GDEs) reduces $CO_2$ by pouring pure $CO_2$ gas onto the catalytic layer of the GDE, delivering it onto the cathode without being dissolved in catholyte in gas-phase electrolysis of $CO_2$. A porous composite electrode, or GDE, is typically made up of polymer-bonded catalyst particles and carbon support. Higher current densities (200–600 mA cm$^{-2}$) may be achieved using GDEs. Furthermore, due to their high porosity and partial hydrophobicity, GDEs create a unique gas–solid–liquid three-phase interface, allowing for a uniform dispersion across the catalytic surface. Because of these characteristics, GDEs are particularly well suited to $CO_2$ electroreduction in the gas phase [13–16]. Janaky's group has reported an important study on the comparison of the operation of electrolyzer cells using different anode materials. The authors showed that while Ir is stable under process conditions, the degradation of Ni leads to a rapid cell failure [17]. The same group has also shed light on other anodic oxygen evolution catalysts that are also detrimental to good GDEs stability during a $CO_2$RR reaction [18]. Electrocatalytic reduction is frequently plagued by high overpotential, poor kinetics, limited product selectivity, and catalyst stability [19–21]. Noble metal electrodes provide great selectivity and stability for $CO_2$ reduction at low over-potential [22–25]. Furthermore, catalysts based on low-cost materials, such as Sn [26–28], Cu [29,30], Co [31–33], and carbon compounds [34–36], are being investigated. CO is the most common result of carbon dioxide reduction reactions ($CO_2$RR), and it is produced by Au, Ag, Zn, and Pd [37]. CO is a significant chemical feedstock for a variety of chemical reactions [16]. The binding energy of *COOH is a critical characteristic for CO generation. The metals bind to *COOH in a sequential manner, generating *CO following dehydration [38–40]. The CO reaction

pathway consumes two electrons and is quite simple [38]. When CO is the major product, there is no need to separate the gaseous CO from the liquid electrolyte because it will spontaneously separate. To this date, we can rank three distinct groups of monometallic catalysts: (1) CO selective metals such as (Au, Ag, Pd, Ga, and Zn) [41,42]; (2) metals that mainly produce HCOOH (e.g., Pb, Cd, Sn, In, and Ti) [43–46]; (3) metals that form hydrocarbons such as $CH_4$ and $C_2H_4$ (e.g., Cu) [47].

Numerous investigations attempting to control the nanostructures of these metals have been known to have high CO production from $CO_2$RR, such as concave rhombic dodecahedral Au nanoparticles with high-index facets [48], TiC-supported Au nanoparticles [49], hexagonal Zn particles [50], electrodeposited Zn dendrites [51], Au electrode with adsorbed $CN^-$ or $Cl^-$ ions [52], Ag nanoparticles with surface-bonded oxygen [53], monodispersed Au or Ag nanoparticles [54,55], ligand-free Au nanoparticles with $<2$ nm [56], and inverse opal Au or Ag thin films [57,58]. Although it is difficult to evaluate their performance due to the differences in reaction circumstances, $CO_2$ to CO conversion typically achieves 90 to 100 percent Faradaic efficiency at a modest overpotential of 0.4 to 0.7 V. Kanan's group [59] has reported remarkable locally enhanced $CO_2$ reduction on gold electrode by studying the influence of bulk defects. These latter appeared to affect the faradaic efficiency and selectivity during $CO_2$ reduction. The same group has also reported the production of formate on $SnO_x$ with a Faradaic efficiency of 58% compared to tin foil only~19%. The recorded current density of 1.8 mA cm$^{-2}$ was achieved at $-0.7$ V vs. RHE [26].

Vasileff et al. [37] discussed and reviewed the surface and interface of copper-based-bimetallic toward $CO_2$ electroreduction. Au has been the most typical group 1 metal alloyed with Cu for the $CO_2$RR because it is a d-block metal with poor hydrogen and oxygen adsorption. Increased Au concentration encouraged CO generation, and the route to $CH_4$ was restricted, according to the Christophe study [60]. CO desorption on Cu sites was aided by the decreased activation energy for CO desorption generated by Au alloying on a mechanistic level. In another study, it was discovered that the composition and nanostructure of Cu–Au nanoparticles influenced catalytic performance, resulting in the selective production of $CH_3OH$ and $C_2H_5OH$ [61].

The Faradaic efficiency (FE) for alcohols in the ideal $Cu_{63.9}Au_{36.1}$ mixture was 28% (including 15.9% for $CH_3OH$), which is 19 times greater than that of pure Cu. This study stated that *CO is a key intermediary in the conversion of $CO_2$ to hydrocarbons and alcohols and that the binding of *CO in this Cu–Au system was likely optimized [62]. Electrochemical results from a comprehensive examination into the effect of the Cu–Au stoichiometric ratio in bimetallic catalysts revealed that alloys with higher Cu content obtained various reduction products, whereas alloys with higher Au content improved CO formation while suppressing other pathways. Cu–Au alloys were shown to enhance CO generation due to their synergistic electrical and geometric effects. The d-band center drops downwards from pure Cu to pure Au, according to density functional theory (DFT) calculations (Figure 1a). As a result, as the Au content increases, the binding strength for *COOH and *CO should decrease, and the generation of CO in Cu–Au systems should follow a monotonic pattern (Figure 1b); however, because of a geometric effect that stabilized *COOH intermediates, *COOH binding was shown to be substantially unaffected. This justifies their experimental outcomes, which show that the $Au_3Cu$ alloy has the highest FE toward CO (Figure 1c), and it gives them a better understanding of the implications of electronic structure and geometric change in bimetallic materials. Furthermore, raising the degree of atomic ordering in Cu–Au alloys was discovered to control the selectivity of $CO_2$ reduction toward CO with a high FE of 80%, which is attributed to the stability *COOH intermediates on compressively strained Au sites [63].

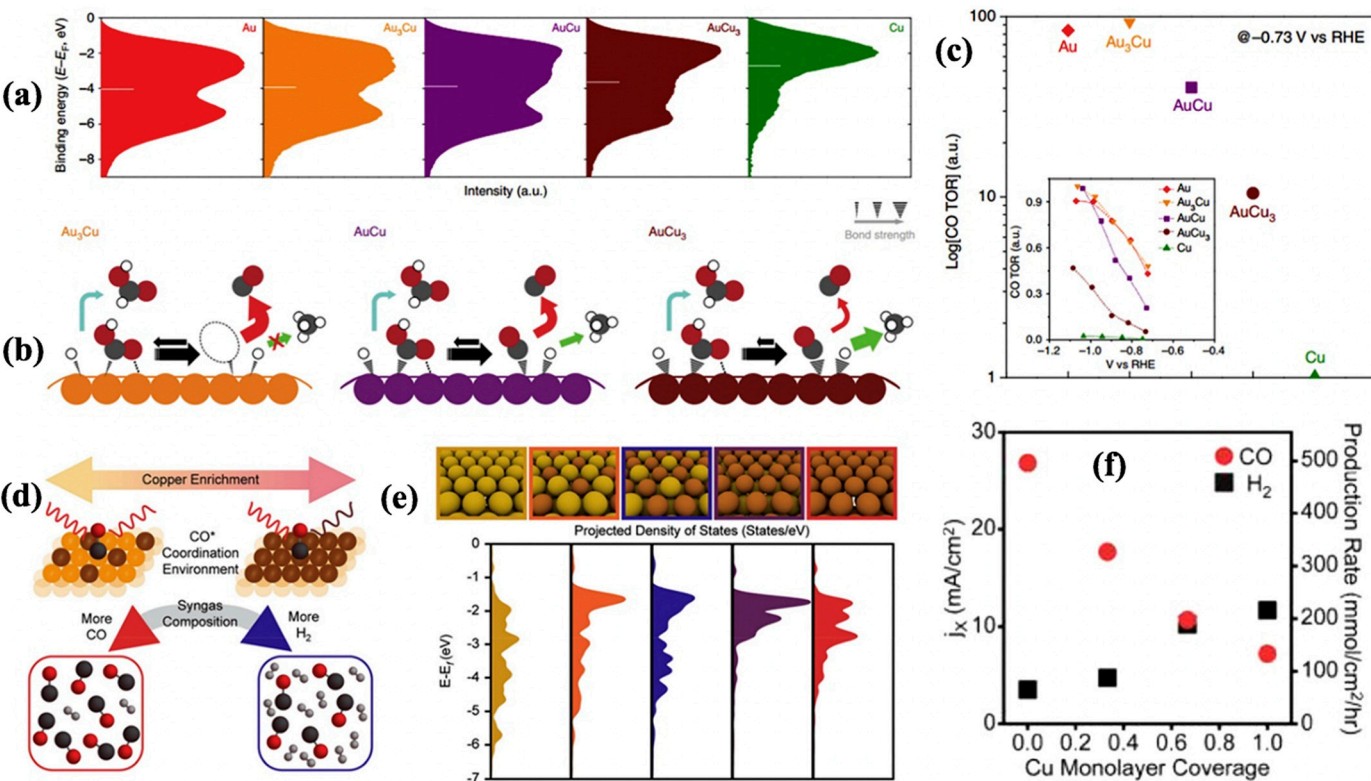

**Figure 1.** Characterization and performance of Cu–Au catalysts for the CO₂RR. (**a–c**) Surface valence-band photoemission spectra of Au–Cu bimetallic nanoparticles (the white bars indicate the d-band centers) (**a**); proposed mechanism for the CO₂RR on the surface of Au–Cu bimetallic nanoparticles (gray, red, and white atoms represent C, O, and H, respectively) (**b**); CO generation rate on various alloy electrocatalysts at a certain overpotential (inset shows relative CO generation rate as a function of the applied potential) (**c**). (Reprinted with permission from Ref. [51]. Copyright 2015 American Chemical Society). (**d–f**) Scheme depicting the relationship between the Cu-enriched Au surface, in situ characterization of CO* coordination, and syngas composition (**d**); calculated d-band electronic states for increasingly Cu-enriched Au surfaces (**e**); partial current densities (left axis) and production rates (right axis) for CO and H₂ as a function of Cu monolayer deposition on Au (**f**). (Reprinted with permission from Ref. [64]. Copyright 2017 American Chemical Society).

By depositing a single layer of Cu with varying coverages, Ross et al. created another form of Cu–Au alloy [64]. They discovered that lower Cu coverage enhanced CO generation, whereas higher Cu coverage promoted $H_2$ evolution. They studied the impact of the Cu/Au ratio on the *CO adsorption strength using in situ Raman microscopy and the vibration of the C-O bond in *CO species as a descriptor. The red shift in vibration of C-O was shown to be linked with bond lengthening due to increased metal contact (Figure 1d). Figure 1e shows that on more Au-dominant surfaces, the projected density of states (DOS) went farther away from the Fermi level, favoring CO generation (DFT calculations). Cu enrichment, on the other hand, improved the adsorption of *H more than it did for *CO. As a result, the degree of Cu enrichment can modify the HER's relative activity to the CO₂RR, allowing for the generation of tunable syngas (Figure 1f). Experiments in a Au–Cu core-shell (Au@Cu) system revealed that seven to eight layers of Cu resulted in greater selectivity for $C_2H_4$, but $CH_4$ generation rose somewhat for 14 or more Cu layers [65]. The computed DFT findings revealed that on terraces, *COH intermediates were preferred over *CHO; but, as *CO coverage rose, *CHO was somewhat favored [66]. As a result, structural and electrical factors that modify the binding of *CO on Au@Cu catalysts have a considerable impact on selectivity and product distribution. In another work, Cu–Au

core-shell nanostructures (Cu@Au) had a higher FE toward CO and had a higher current density than polycrystalline Cu [67].

Contemporary research has found that oxide-derived metal electrodes outperform virgin metal electrodes in terms of catalytic performance [68]. The most typical method for making oxide-derived electrocatalysts is to oxidize the metal before reducing it to its original metallic form. At the same overpotential, oxide-derived Cu generated more $C_2$ products of $C_2H_4$, $C_2H_6$, and ethanol than electropolished Cu [68–70]. Cl ion-adsorbed oxide-derived Cu [71] yielded $C_3$ and $C_4$ products of $C_3H_7OH$, $C_3H_6$, $C_3H_8$, and $C_4H_{10}$. For the synthesis of formic acid, oxide-derived Sn had a substantially greater current density and Faraday efficiency than a pure Sn electrode [72]. As shown in Figure 2a, oxygen-derived Au [73] or Ag [74] had a higher FE for CO generation, ranging from 90 to 100 percent at only 0.3 V overpotential.

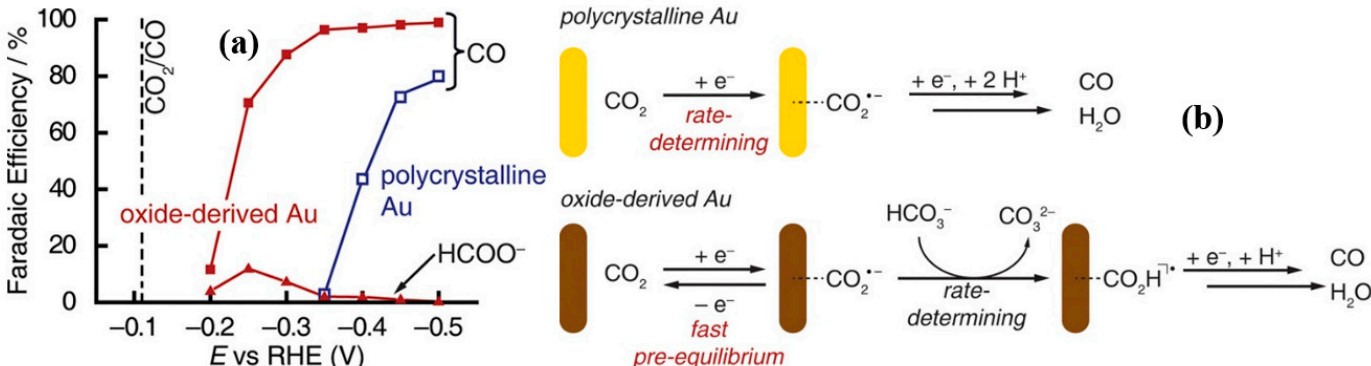

**Figure 2.** (**a**): FEs for CO and $HCO_2^-$ production on oxide-derived Au and polycrystalline Au electrodes at various potentials between $-0.2$ and $-0.5$ V in 0.5 M $NaHCO_3$, pH 7.2. Dashed line indicates the CO equilibrium potential. (**b**): suggested mechanisms for $CO_2$ reduction to CO on polycrystalline Au and oxide-derived Au (Reprinted with permission from Ref. [74]. Copyright 2016 Wiley-VCH).

The origin of the increase in oxide-derived metals for $CO_2RR$ has been a point of conflict. The source of the strengthening has been proposed to be nanostructures produced in the catalysts or residual subsurface oxygen. After oxidation-reduction cycling, nanostructured surfaces with rich grain boundaries were formed [75,76]. More defect sites with greater binding energy to *CO and higher local pH were found on the induced surface, which improved selectivity toward $CO_2RR$ while reducing HER [76–78]. Despite the $CO_2RR$'s extremely decreased circumstances, it was also hypothesized that some oxygen persisted in the subsurface. As demonstrated in Figure 2b [79,80], ambient pressure XPS, in situ electron energy loss spectroscopy examinations, and DFT simulations revealed that oxygen assisted in the early activation of $CO_2$ on the surface; however, a conflicting conclusion was also published, claiming that under the $CO_2RR$ situation, the remaining oxide was highly unstable and the amount of oxygen was insignificant [81].

Commercial $Cu_2O$ and $Cu_2O$–ZnO mixes coated on carbon sheets have been used by Albo et al. [82] to make gas-diffusion electrodes, which are tested in a filter-press electrochemical cell for continuous $CO_2$ gas-phase electroreduction (Figure 3). The operation mostly yielded methanol, with minor amounts of ethanol and n-propanol. The investigation uses a 0.5 M $KHCO_3$ aqueous solution to measure critical variables affecting the electroreduction process: current density (j = 10–40 mA $cm^{-2}$), electrolyte flow/area ratio (Qe/A = 1–3 mL $min^{-1}$ $cm^{-2}$), and $CO_2$ gas flow/area ratio (Qg/A = 10–40 mL $min^{-1}$ $cm^{-2}$). At an applied potential of 1.39 and 1.16 V vs. Ag/AgCl, respectively, the greatest $CO_2$ conversion efficiency to liquid-phase products was 54.8 percent and 31.4 percent for $Cu_2O$ and $Cu_2O$/ZnO-based electrodes.

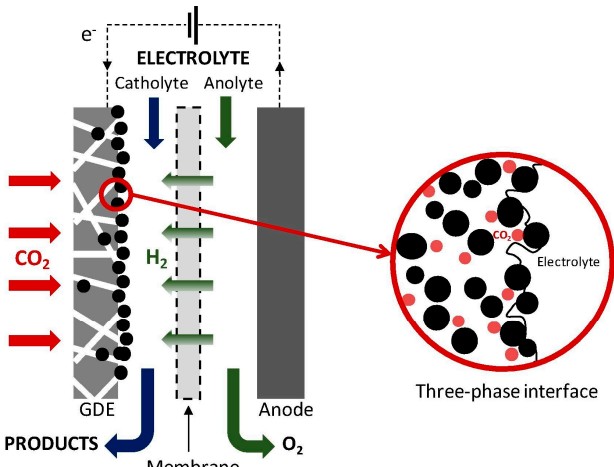

**Figure 3.** Representation diagram of the electrolytic cell configuration for the electroreduction of $CO_2$ supplied directly from the gas phase. In this study, the filter-press electrochemical system possesses three inputs (catholyte, anolyte, and $CO_2$ separately) and two outputs (catholyte–$CO_2$ and anolyte) for the electroreduction of $CO_2$ in gas phase. (Reprinted with permission from Ref. [82]. Copyright 2016 Elsevier).

## 2.2. Single-Atom Catalysts

Single-atom catalysts (SAC) are metal catalysts that are atomically scattered on the surface of a support. They have unusual selectivity and have extremely different electronic structures and adsorption patterns of reactants and intermediates [83]. Electrochemical $CO_2RR$ [84] has been studied using SACs. As shown in Figure 4 [84], Ni single atoms on N-doped graphene can selectively catalyze $CO_2RR$ and produce CO. Various metal atoms with quite different d-band structures, such as Fe, Co, Mn, and Cu, have demonstrated varying selectivity. The electronic structure and binding energy of important carbon intermediates have been found to be affected by the coordination environment around the Ni single atom [85,86]. Cu Ion-O vacancy pairs formed when Cu was atomically doped into $CeO_2$, and the oxygen defect sites of $CeO_2$ governed copper's oxidation state. $CO_2RR$ was catalyzed by a single atomic Cu, which produced $CH_4$ with a 58 percent Faraday efficiency [87]. With the right metal atoms and supports, DFT calculations predict that the SAC structure can suppress HER while promoting $CO_2RR$ [88,89]. Computational methods have also suggested single-atom alloys, in which a single-atomic metal is located on another metal surface, as effective $CO_2RR$ catalysts [90].

Pan and coworkers have developed a $Co-N_5$ single atom through pyrolysis and wt impregnation method for the electrocatalytic reduction of $CO_2$ to Co, reaching a Faradaic efficiency of 99% at $-0.73$ V vs. RHE at a current density of 6.2 mA $cm^{-2}$. The Co SAC was stable for 10 h of electrolysis [33]. Another Co single atom ($Co-N_2$) has been reported by [91] from the pyrolysis of Co/Zn MOF; this electrocatalyst achieved a Faradaic efficiency for CO2 reduction of 94% at $-0.63$ V vs. RHE and a current density of 18.1 2 mA $cm^{-2}$. This catalyst had a steady-state stability of 60 h. Iron single atoms have also been reported for $CO_2$ electroreduction; for instance, Pan and coworkers [92] have recently reported Fe-N-C prepared from Fe salt, Zn salt, and 2-methylimidazole. The prepared material was evaluated for $CO_2$-to-CO with a selectivity of 93% at $-0.7$ V vs. RHE and a current density of 2.8 mA $cm^{-2}$. This Fe SAC showed a steady-state CO evolution for 20 h. Zhang et al. [93] have synthesized an Fe/NG SAC electrocatalyst using Lyophilization and reduction of the precursors (Graphene oxide and Iron salt). The Fe/NG SAC yielded 80% Faradaic efficiency for CO and stability of 10 h.

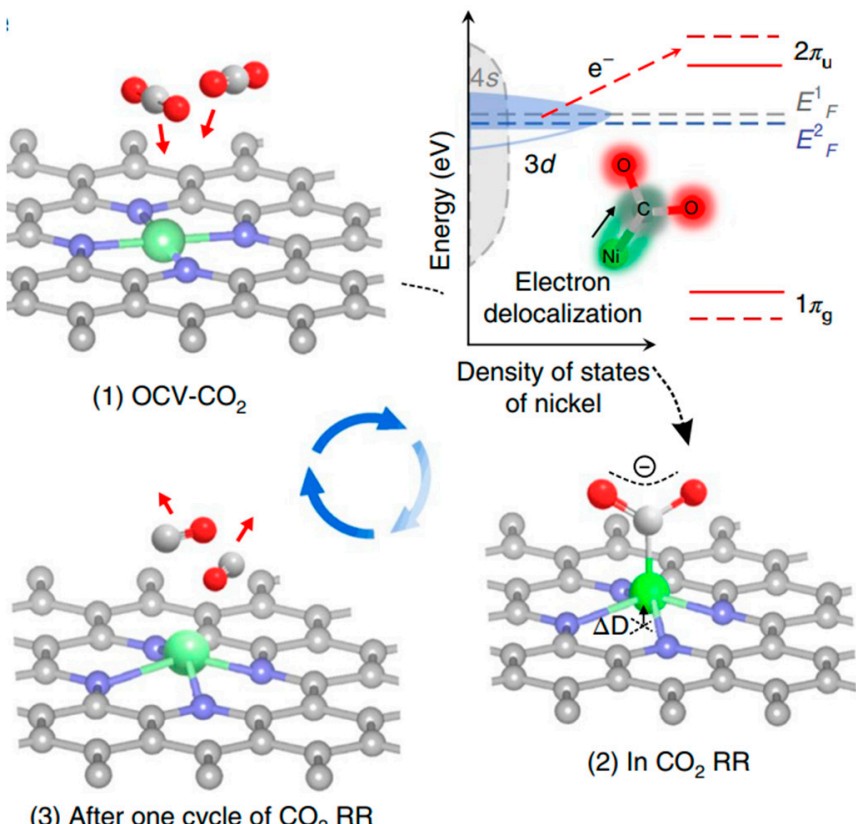

**Figure 4.** Structural evolution of Ni single-atom sites on graphene during $CO_2$RR. $\Delta$D in shows the displacement of Ni atom out of plane resulting from electron transfer from Ni atom to $CO_2$. The upper-right schematic shows the activation processes for $CO_2$ molecules on the Ni(i) site. A valence band structure, similar to metallic nickel, was used to simplify the schematic illustration. The red arrow represents the electron transfer from the Ni(i) to adsorbed $CO_2$. $E_F^1$ and $E_F^2$ are Fermi levels of A-Ni-NG before and after formation of Ni-$CO_2$, respectively. $1\pi_g$ and $2\pi_u$ are $CO_2$ molecular orbitals. (Reprinted with permission from Ref. [84]. Copyright 2018 SpringerNature).

In a $CO_2$-saturated 0.1 M $KHCO_3$ solution, the electrocatalytic performance of Cu-$CeO_2$ nanorod samples was evaluated [87]. CV curves of Cu-$CeO_2$-4% nanorods, undoped $CeO_2$ nanorods, and Cu nanoparticles vs. a reversible hydrogen electrode (RHE, Figure 5a) were measured in a potential window between −0.2 and −1.8 V. Because $CeO_2$ has a lower electrical conductivity than Cu, the Cu-$CeO_2$-4% nanorods (red curve) have a lower total current density ($J_{tol}$) than pure Cu (blue curve), although they are still much greater than $CeO_2$ (black curve). Figure 5b–d show the electrocatalytic reduction products of these three catalysts as determined by in-line gas chromatography (GC) and $^1$H nuclear magnetic resonance (NMR). Cu-$CeO_2$ nanorods had a much smaller quantity of $H_2$ side products from water reduction (gray columns in Figure 5b–d) and a significantly higher number of electrocatalytic $CO_2$ reduction products than Cu nanoparticles and undoped $CeO_2$ nanorods, showing strong $CO_2$ reduction activity on the Cu-doped $CeO_2$ electrocatalysts. The current densities for all deep reduction products ($j_{drp}$) were computed and plotted by multiplying $j_{total}$ and the corresponding FEs for all deep reduction products (i.e., $CO_2$ reduction products excluding CO and $HCOO^-$) (Figure 5b–d, red *y*-axis on the right). At −1.6, −1.8, and −2.0 V versus RHE, the $j_{drp}$ of the Cu-$CeO_2$-4% nanorods reaches 40, 70, and 90 mA cm$^{-2}$ (based on geometric surface area), which was many times greater than the pure Cu and undoped $CeO_2$ samples.

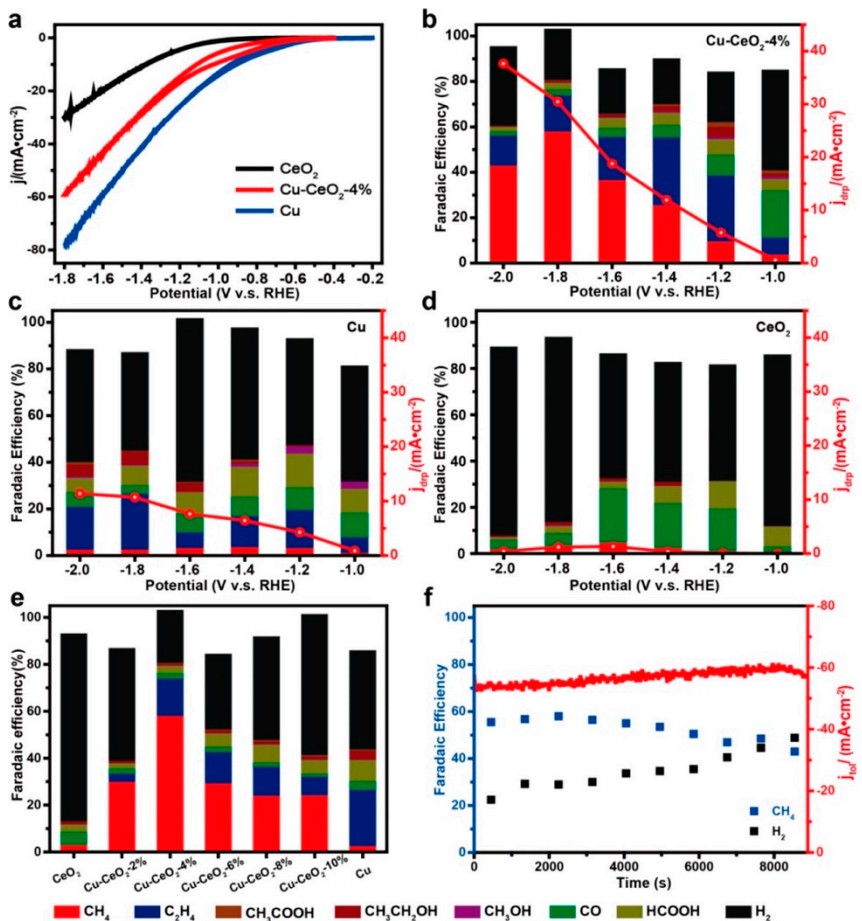

**Figure 5.** Electrochemical $CO_2$ reduction performance. (**a**) Cyclic voltammetry curves for Cu-$CeO_2$, $CeO_2$, and Cu. (**b**−**d**) Faradaic efficiencies (bars on the left $y$−axis) and deep reduction product current density ($j_{drp}$, red curves on the right $y$−axis) of (**b**) Cu−$CeO_2$−4%, (**c**) pure Cu, and (**d**) undoped $CeO_2$ at various overpotentials. The deep reduction products were the first five products in the legends at the bottom, marked with a red line. (**e**) A comparison of the Faradaic efficiency of samples with varying levels of Cu doping. (**f**) Stability of $FE_{CH4}$ (blue squares) and $FE_{H2}$ (black squares) on the left $y$−axis. Right $y$−axis: total current density ($j_{total}$) of Cu−$CeO_2$−4% at −1.8 V (red curves, right $y$−axis). (Reprinted with permission from Ref. [87]. Copyright 2018 American Chemical Society).

### 2.3. Carbon-Based Catalysts

Carbon-based materials are being investigated as potential electrocatalysts for electrochemical processes such as the oxygen reduction reaction (ORR), the oxygen evolution reaction (OER), and the hydrogen evolution reaction (HER) [94,95]. Carbon materials provide a number of benefits for electrochemical applications, including the capacity to be easily transformed into different sizes and forms [96]. Well-developed material science techniques may be used to create zero-dimensional carbon dots, graphene quantum dots, one-dimensional carbon nanotubes, two-dimensional graphene, and three-dimensional graphene aerogel [21]. Carbon materials have strong chemical and mechanical stability, as well as high conductivity and surface area.

$CO_2RR$ is essentially inert to pure carbon compounds. Electrocatalytic activity is considerably improved when heteroatoms such as N are doped in the carbon matrix. $CO_2RR$ considers negatively charged N sites to be active sites [97,98]. N-doping the catalyst creates a Lewis base site, which helps to stabilize $CO_2$ [98]. N-doped carbon nanotubes [99,100], N-doped graphene [101], and N-doped graphene quantum dots [35] are among the carbon materials that have been reported for $CO_2RR$ through the two-

electron route. Acetate and formate with N-sp³C active sites were synthesized by N-doped diamonds [102]. $CO_2RR$ has also been performed with other dopants, such as S or B [103,104].

Pan et al. [33] used an N-coordination technique to build a stable $CO_2$ reduction reaction electrocatalyst with an atomically distributed Co–$N_5$ site anchored to hollow N-doped porous carbon spheres made from polymers. The synthesis steps are depicted in Figure 6. The authors use a modified Stöber technique [105] to make core@shell $SiO_2$@melamine-resorcinol-formaldehyde polymer spheres (MRFPSs). Then, by pyrolyzing $SiO_2$@MRFPSs at 700 °C under Ar, $SiO_2$@N-doped porous carbon spheres were produced. The HNPCSs were created after etching the silica core with HF. Finally, the $CoN_5$/HNPCSs catalyst was developed by establishing a coordination connection between Co and N. The catalyst has a strong selectivity for $CO_2$ reduction, with a CO Faradaic efficiency ($FE_{CO}$) of more than 90% throughout a wide potential range of −0.57 to −0.88 V (the $FE_{CO}$ approached 99 percent at −0.73 and −0.79 V). After electrolyzing for 10 h, the CO current density and $FE_{CO}$ remained practically constant, demonstrating excellent stability. Experiments and density functional theory simulations demonstrate that the single-atom Co–$N_5$ site is the primary active center for simultaneous $CO_2$ activation, fast production of the critical intermediate COOH*, and CO desorption.

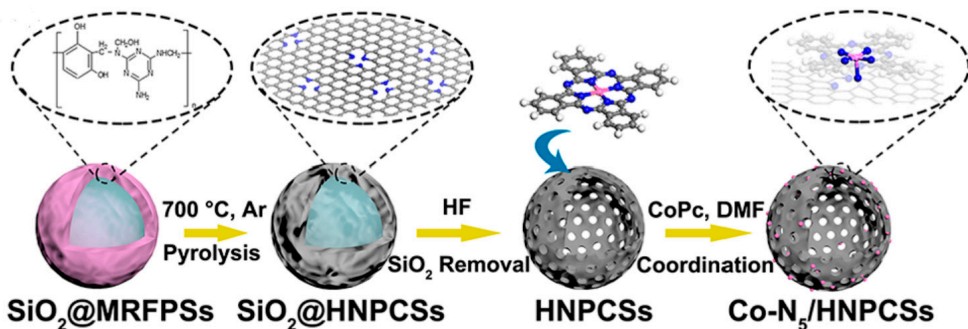

**Figure 6.** Schematic illustration of catalyst preparation. The core@shell $SiO_2$@melamine-resorcinol-formaldehyde polymer spheres (MRFPSs) were first synthesized. Then, $SiO_2$@N-doped porous carbon spheres were obtained by pyrolysis of $SiO_2$@MRFPSs at 700 °C under Ar. After etching the silica core with HF, the HNPCSs were obtained. Finally, the Co−$N_5$/HNPCSs catalyst was prepared through constructing coordination interaction between Co and N. (Reprinted with permission from Ref. [33]. Copyright 2018 American Chemical Society).

Carbon compounds doped with nitrogen are effective $CO_2$ electroreduction catalysts [21,106]. In ionic liquids, nitrogen-doped carbon nanofibers showed a modest overpotential for $CO_2$ reduction [36]. Although the product selectivity of each was very poor, the N-doped graphene quantum dot generated multi-carbon compounds such as ethanol, acetate, and n-propanol [35]. In aqueous conditions, nitrogen-doped mesoporous carbon was recently employed to convert $CO_2$ to ethanol with great efficiency and selectivity (77%) [34]. Other N-doped materials for $CO_2$ reduction reactions have been reported, including N-doped graphene [107,108], and carbon [109,110].

Because of their unique electrical and geometric properties [111], N-doped carbon nanotubes have attracted a lot of interest in $CO_2$ reduction and other electrocatalysis [112]. NCNTs modified with polyethylenimine may convert $CO_2$ to formate in aqueous environments with good selectivity (87%) and current density [113]. NCNT arrays produced by chemical vapor deposition (CVD) demonstrated good selectivity for CO of 80% at −0.26 V overpotential [99]. The high density of pyridinic N sites (27 percent of all Ns) in this NCNT array [100] was attributable to its superior activity. When compared to a reversible hydrogen electrode (RHE) with a pyridinic N concentration of 32%, NCNTs catalysts produced by calcination of polymers may obtain a maximum current efficiency of 90% for CO production at a potential of 0.9 V [98]. In comparison to noble-metal catalysts,

however, such NCNT catalysts have a low CO current density. For more effective catalysts, increasing the N concentration is crucial.

Pyridinic N is the most significant catalytic site for carbon dioxide reduction reactions ($CO_2$RR) among nitrogen-doped carbon materials. Graphitic N defects have a lower ability to bind $CO_2$, while pyrrolic N defects have little to no effect on $CO_2$RR activity; however, the atomic abundance of pyridinic N in most catalysts was low, accounting for just around 30% of all N atoms, and only a few materials could achieve a pyridinic N content of over 60%. Increasing the number of pyridinic N defects is therefore crucial. Pyrolysis is an effective method for producing high-N-content NCNTs [101,108,109].

Pyrolysis of electrospun nanofiber mats of heteroatomic polyacrylonitrile (PAN) polymer yielded a low-cost, metal-free carbon nanofiber (CNF) catalyst for $CO_2$ reduction [36]. This catalyst's heteroatomic structure was intended to make use of nitrogen atoms already present in the precursor's backbone (PAN). CNF catalysts include two electrochemically active species: pyridinic nitrogen and positively charged carbon atoms. To balance out the pyridinic N's high negative charge density, the adjacent carbon atom has a higher positive charge density and is an oxidized carbon. If the fascinating $CO_2$ conversion seen for CNFs is linked to nitrogen functional groups, the composition of nitrogen atoms should alter dramatically following the experiment. The authors [36] evaluated the change in the N atom configuration of the CNFs catalyst by recording high-resolution N1s spectra before and after the 9 h electrochemical reaction (Figure 7). In CNFs, Figure 7a shows the presence of three primary nitrogen species: pyridinic (B.E. ~398.5 eV), quaternary (B.E. ~401.1 eV), and nitrogen oxides (B.E. ~402.2 eV). The quantitative studies show that pyridinic nitrogen makes up 25.8%, quaternary nitrogen makes up 36.7 percent, and N-oxide makes up 37.5 percent of the total nitrogen. An extra prominent N peak (pyridonic nitrogen) was identified at B.E. (400.1 eV) in the N1s spectra (Figure 7b) collected for CNFs following the 9 h reaction. According to the authors, the size of the N-oxide peak reduces dramatically (37.5–10%), showing that N-oxides are transformed into pyridonic nitrogen (peak area varies from 0 to 37%). Though, the CNF catalysts' electrochemical activity stays unaltered, indicating that N-oxides were not involved in the $CO_2$ conversion process. Surprisingly, following the 9-h trials, the strength of the pyridinic (very active) and quaternary nitrogen (less active) peaks stays practically unchanged. Furthermore, the authors speculated that there might be two alternative processes if pyridinic nitrogen was directly involved in the reaction. First, pyridinic nitrogen may be permanently protonated, resulting in quaternary nitrogen [114], which would increase the peak area of quaternary nitrogen since the B.E. for protonated nitrogen is comparable to that of quaternary (401.3 eV) [115]. Instead of getting irreversibly protonated, pyridinic nitrogen might weakly attach to $CO_2$ species in a way similar to pyridine reduction, resulting in pyridinic nitrogen conversion to pyridinic species [114]. In both situations, the peak area of the pyridinic nitrogen peak would have been reduced (after tests), but it remains the same (25.8%), demonstrating that such reactions do not occur in the system. As a result, N1s spectra studies force authors to consider that positively charged carbon atoms lead to the electrochemical reduction of $CO_2$. This is further supported by the fact that nitrogen-free carbon atom catalysts (such as graphite) have a very low $CO_2$ reduction current density.

Based on theoretical [116,117] and experimental [118–120] investigations, the oxidized carbon atoms can function as excellent catalysts for reduction reactions due to their high atomic charge and spin density. The naturally oxidized carbon atoms can be decreased at first by the redox cycling mechanism. The intermediate complex [EMIM–$CO_2$] adsorbs on reduced carbon atoms and reoxidizes them to their original form, yielding CO as a product in the second step (Figure 7e).

Wu and coworkers [101] employed a CVD method to create microporous graphene foams, which were subsequently doped with nitrogen (N) using graphitic carbon nitride (g-$C_3N_4$). In terms of CO generation, the resulting N-graphene was both active and selective, with a nitrogen content of 6.5 percent. At −0.58 V vs. the reversible hydrogen electrode (RHE), which is an overpotential of −0.47 V. Formate production was also re-

ported, although at a low FE of 3% (at −0.58 V), implying that the $CO_2RR$ on this catalyst material follows a $2e^-$ route.

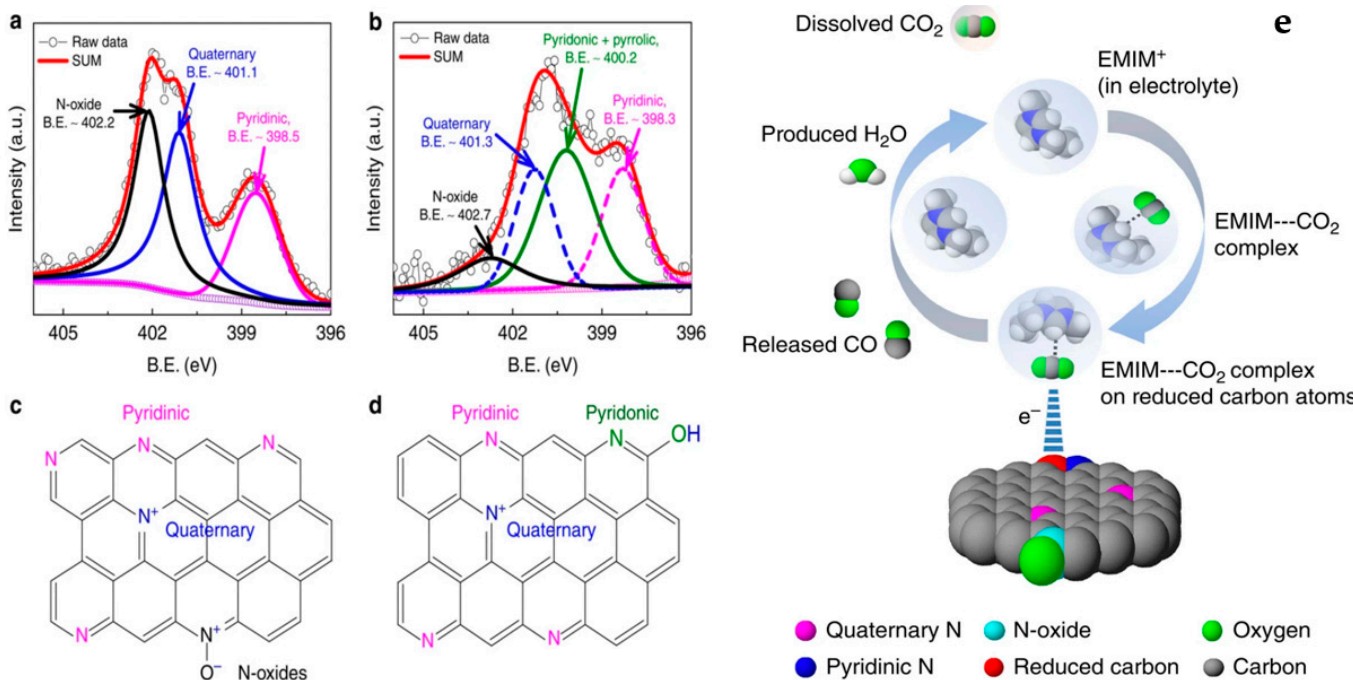

**Figure 7.** (**a–d**) Evolution of nitrogen atomic nature in CNFs by XPS. (**a**) Deconvoluted N1s spectra for CNFs before and (**b**) after electrochemical experiments. In used catalysts, CNFs N1s spectra, N-oxide type of nitrogen content reduced radically and new peak (green solid line) at 400.2 eV (pyridonic N) appears. (**c,d**) The corresponding atomic structure on the basis of XPS analysis. (**e**): $CO_2$ reduction mechanism schematic diagram. The $CO_2$ reduction reaction takes place in three steps: (1) an intermediate (EMIM–$CO_2$ complex) formation, (2) adsorption of EMIM–$CO_2$ complex on the reduced carbon atoms, and (3) CO formation. (Reprinted with permission from Ref. [36]. Copyright 2013 SpringerNature).

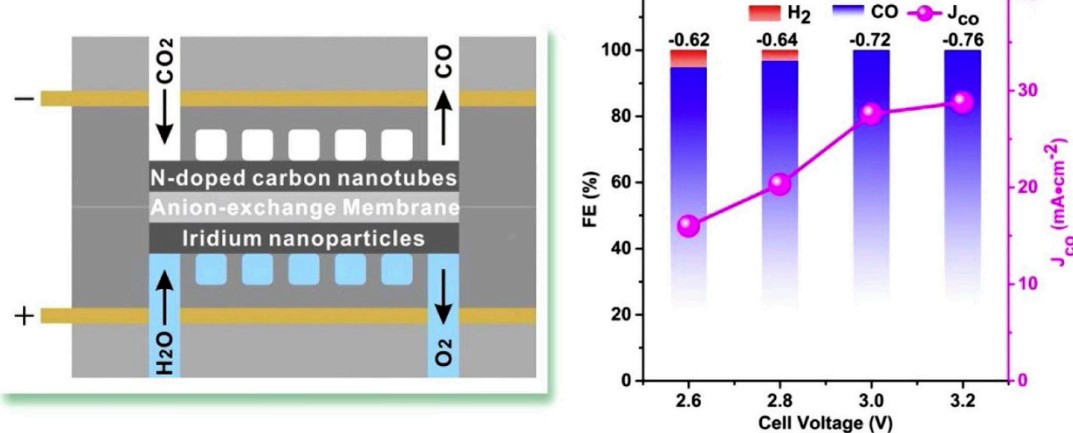

**Figure 8.** Gas−phase electrolysis in a flow cell and gas-phase $CO_2$ electrolysis. FE of CO (blue) and $H_2$ (red) vs. cell voltage (left axis) and partial current density of CO vs. cell voltage (right axis), values above the column are cathode potentials vs. normal hydrogen electrode (NHE) (Reprinted with permission from Ref. [121]. Copyright 2019 Elsevier).

To boost the concentration of pyridine N, phenanthroline was utilized as a precursor [121]. The effects of N doping and atomic configurations on activity were studied, and catalytic active sites were discovered. The mechanism of $CO_2RR$ on NCNTs was also

postulated. By pyrolysis of a phenathroline heterocycle precursor, nitrogen-doped carbon nanotubes (NCNTs) with a high concentration of pyridinic N sites (62.3 percent) were created, and they can convert $CO_2$ to CO with great selectivity and stability. Between 0.6 and 0.9 V versus the reversible hydrogen electrode (RHE), the Faradaic efficiency of CO was maintained at >94.5 percent, and the CO current density was as high as −20.2 mA cm$^{-2}$ (Figure 8). Furthermore, after 40 h of electrolysis at −0.8 V, the CO faradic efficiency remained stable at 95% (Figure 8). The remarkable performance was attributed to the large quantities of pyridine N sites in NCNTs, which serve as catalytic active sites. Furthermore, gas-phase $CO_2$ electrolysis demonstrated approximately 100% Faradic efficiency for CO (Figure 8), implying that the NCNT can optimize $CO_2$ reduction efficiency while entirely suppressing hydrogen evolution.

Researchers coupled a high inherent defect density acquired by adjusting the size and shape of carbon nanostructures at the nanometer scale with foreign N-doping to create an improved metal-free catalyst for the electroreduction of $CO_2$ to value-added compounds (Figure 9) [35]. The end product, N-doped graphene quantum dots (NGQDs), has a much higher density of N-doping defects at edge locations. These NGQDs have high activity for electrochemical $CO_2$ reduction, as evidenced by high reduction current densities at low overpotentials, and, more importantly, they preferentially produce multi-carbon hydrocarbons and oxygenates, particularly the $C_2$ products ethylene ($C_2H_4$) and ethanol ($C_2H_5OH$), at FEs comparable to those obtained with Cu nanoparticle-based catalysts.

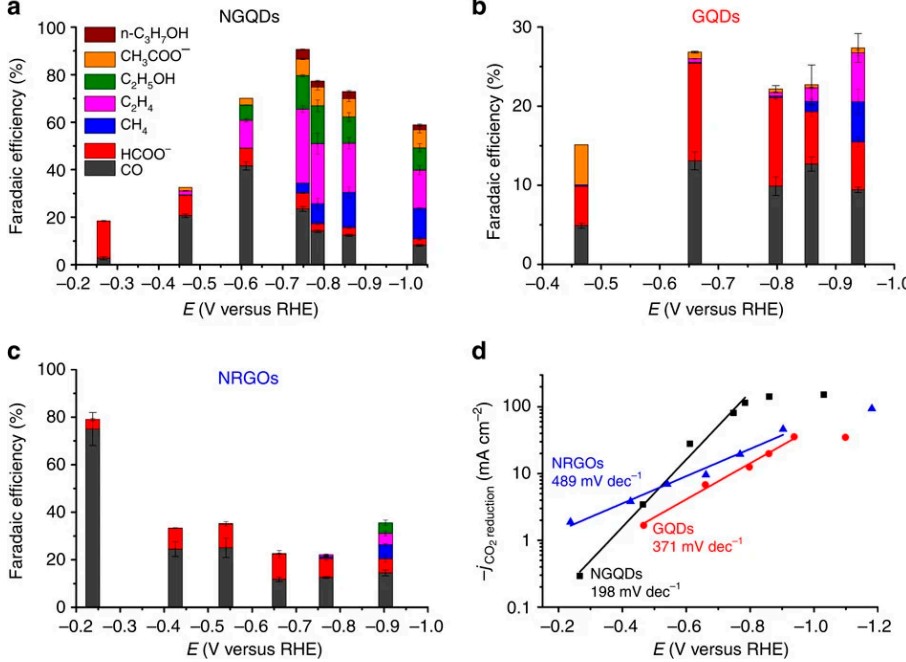

**Figure 9.** Electrocatalytic activity of carbon nanostructures towards $CO_2$ reduction. (**a**) FEs of carbon monoxide (CO), methane ($CH_4$), ethylene ($C_2H_4$), formate ($HCOO^-$), ethanol (EtOH), acetate ($AcO^-$) and n-propanol (n-PrOH) at various applied cathodic potential for NGQDs. (**b**) FE of $CO_2$ reduction products for pristine GQDs. (**c**) Selectivity to $CO_2$ reduction products for NRGOs. (**d**) Tafel plots of partial current density of $CO_2$ reduction versus applied cathodic potential for three nanostructured carbon catalysts. The error bar represents the s.d. of three separate measurements for an electrode. (Reprinted with permission from Ref. [35]. Copyright 2016 SpringerNature).

### 2.4. Porphyrins, Covalent, and Metal-Organic Framework Catalysts

Porphyrins and other organometallic compounds have long been explored for the $CO_2$RR and were among the first materials to solve the problems associated with metal electrodes [1,122,123]. In our recent study, we have elucidated the working state of iron porphyrin for the $CO_2$-to-CO in an aqueous medium [124]. Using operando UV-

spectroscopy and X-ray absorption near-edge structure spectra, we have confirmed that during the cathodic reaction, the Fe(II) species acts as catalytic sites that accommodate CO as Fe(II)–CO adducts. DFT studies have confirmed and pointed out that the ligand $[Fe(II)F_{20}(TPP\bullet)]-$ prevails in the catalytic cycle prior to the rate-controlling step.

Metal-functionalized porphyrin materials that are employed as $CO_2RR$ electrocatalysts are typically synthesized as homogeneous molecular catalysts. As a consequence, a variety of porphyrins may be functionalized with a variety of transition metal centers (Co, Fe, Zn, Cu, etc.) and the resulting electrocatalysts have well-defined active centers that can be carefully adjusted for high activity and selectivity towards the $CO_2RR$ [1,125–129].

The active centers of heterogeneous electrocatalysts are typically difficult to characterize, making performance optimization difficult. Heterogeneous electrocatalysts, on the other hand, provide stability in electrocatalytic function, particularly in aqueous conditions, which is critical for their practical use. In general, however, these molecular catalysts are unstable and degenerate after just a few catalytic cycles, in addition to catalyst–electrolyte separation difficulties [128]. In order to synergistically boost the stability and effectiveness of homogeneous porphyrin-based catalysts, such as weak Lewis and Brönsted acids, a co-catalyst is usually used in solution with them [130–132]. The production of Fe porphyrin dimers with carefully adjusted Fe center spacing is a new example that addresses some of these challenges [128]. Without the use of co-catalysts, the presence of two metal centers resulted in coordinated stabilization and binding of $CO_2$ molecules, which boosted activity and stability.

Transition metal porphyrin and other organometallic materials have recently been synthesized and constructed as metal–organic frameworks (MOFs) and covalent organic frameworks (COFs) and functionalized on other nanostructured supports such as graphene and MWCNTs in an effort to combine the exemplary features of both homogeneous and heterogeneous electrocatalysts [125–127,133]. The porphyrin-based materials are typically prepared using solvothermal or hydrothermal methods, and the resulting materials are then placed on conductive substrates to generate $CO_2RR$ electrodes. The structure of the prophyrins and their well-defined active metal centers are preserved since the electrocatalysts are built without high-temperature stages. Deposition on conductive substrates enhances the stability and electron transport to these active sites, resulting in an improved electrocatalytic activity.

Weng et al. [129] describe copper (II)-5,10,15,20-tetrakis (2,6-dihydroxyphenyl) porphyrin (PorCu), a novel copper-porphyrin complex with unique catalytic characteristics for $CO_2$ reduction in neutral aqueous conditions. For electrochemical $CO_2$ reduction to hydrocarbons, the PorCu catalyst exhibits great activity and selectivity (methane and ethylene). The catalyst converted $CO_2$ to hydrocarbons with a Faradaic efficiency of 44 percent at a mass loading of 0.25 mg/cm$^2$ and an electrochemical potential of $-0.976$ V vs. the reversible hydrogen electrode (RHE), significantly inhibiting the other $CO_2$ reduction routes. Under the same circumstances, an ultrahigh geometric current density of 21 mA/cm$^2$ was reached for simultaneous methane and ethylene production, resulting in turnover frequencies (TOFs) of 4.3 methane and 1.8 ethylene molecules. site$^{-1}$ s$^{-1}$. The built-in hydroxyl groups on the porphyrin ligand, as well as the oxidation state of the Cu center, were both key elements leading to the higher catalytic activity, according to this research.

Iron porphyrin monomers have been reported to catalyze the electrochemical $CO_2$ reduction to CO in DMF/tetraalkylammonium salts as a supporting electrolyte, with high selectivity at the electro-generated $[Fe^{I}(por)]^{2-}$ (conventionally described as $[Fe^{0}(por)]^{2-}$ species [134] among the non-precious metal-based $2e^-/2H^+$ coupled $CO_2$ electrochemical reduction catalysts. These catalysts, on the other hand, degrade after just a few catalytic cycles [135]. The presence of Lewis acids such as $Mg^{2+}$ and $Ca^{2+}$ [132,136], or weak Brönsted acids such as trifluoroethanol and 1-propanol [130], boosts their catalytic efficiency and stability via a push–pull process where the electro-generated electron-rich $[Fe^{0}(por)]^{2-}$ increases their catalytic efficiency and stability. The electron-deficient synergist Lewis or Brönsted acid stimulates the breaking of one of the C–O bonds by pushing an electron pair

to the $CO_2$ molecule [131]. Due to the large local concentration of protons associated with the phenolic hydroxy substituents, modification of the iron tetraphenylporphyrin monomer, FeTPP, with phenolic hydroxy groups at all phenyl group ortho positions improves its activity and stability for $CO_2$ electro-reduction to CO [137]. Carbon monoxide dehydrogenase (CODH) is a metalloenzyme with a Ni–Fe dinuclear complex at its active core that facilitates the selective conversion of $CO_2$ to CO [138]. The electrochemical process supported by CODH produces catalytic $CO_2$ reduction with a very low overpotential [139]. As a result, dinuclear catalysts are excellent candidates for $CO_2$ reduction catalysis. Naruta groups [128] previously documented the utilization of multiple cofacial porphyrin dimers as ligands for retaining two manganese ions with an appropriate Mn–Mn separation distance (3.7–6.2 Å) to facilitate water oxidation to oxygen or $H_2O_2$ disproportionation [140]. The utilization of a dimeric combination of two iron ions as bio-inspired catalysts was described for the first time by Naruta team [128]. In a DMF/10 percent $H_2O$ solution, a cofacial iron tetraphenyl porphyrin dimer, o-$Fe_2$DTPP (Figure 10), effectively and selectively catalyzes the electrochemical reduction of $CO_2$ to CO. The activity of the 1,3-phenylene bridged iron porphyrin dimer (FeTPP) was compared to that of the comparable iron porphyrin monomer (FeTPP) (m-$Fe_2$DTPP). The CVs of o-$Fe_2$DTPP (0.5 mM) in a DMF/10% $H_2O$ solution containing TBAPF$_6$ (TBAPF$_6$ = tetra-n-butylammonium hexafluorophosphate, 0.1 M) saturated with Ar gas (blue line) or $CO_2$ gas (red line) are shown in Figure 10b. The detection of a significant catalytic current in the presence of $CO_2$ gas was the most intriguing discovery, showing electrocatalytic $CO_2$ reduction enhanced by o-$Fe_2$DTPP. The emergence of the catalytic peak over the $Fe^+$–$Fe^+$/$Fe^+$–$Fe^0$ redox couple under Ar at $-1.48$/$-1.46$ V vs. NHE (later, all potentials are given against NHE unless otherwise mentioned) shows the beginning of the catalytic process after the $Fe^+$–$Fe^0$ porphyrin species was electro-generated. Figure 10c depicts the temporal sequence of the products' appearance. Without the catalyst, $H_2$ is the only product produced with 99 percent Faradic efficiency and a low average current density of $-0.1$ mA cm$^{-2}$. In the presence of o-$Fe_2$DTPP, however, a considerable charge, Q = 41.4 C, was burned at an average current density of $-1.15$ mA cm$^{-2}$ across a 10 h electrolysis process, with the concurrent generation of CO (88 percent Faradic efficiency) and $H_2$ (12 percent Faradic efficiency). When the quantity of $H_2$ created during the control experiment is subtracted, the dimer catalyzes the $CO_2$ reduction to CO with 95% Faradic efficiency. The present density–time profile revealed no reduction over the previous ten hours.

A study describes the electrochemical reduction of $CO_2$ to CO and methane, as well as trace quantities of HCOOH and methanol, using a simple Co protoporphyrin molecular catalyst immobilized on a pyrolytic graphite (PG) electrode in a completely aqueous electrolyte solution [127]. Previous research utilizing immobilized Co porphyrins or Co phthalocyanines demonstrated that Co-based catalysts might reach a high FE towards CO, which is very sensitive to pH and potential [141–143]. Immobilized Co-based porphyrins are excellent $CO_2$ reduction electrocatalysts, according to Shen et al. [127], and can produce multi-electron products such as methane and methanol. More importantly, this research highlights the importance of pH in directing catalytic activity and selectivity towards CO and $CH_4$, particularly in the absence of coordinating anions in the pH range of 1–3. This great sensitivity to pH is explained by a mechanism that emphasizes the critical significance of the first electron transfer in electrochemically activating $CO_2$. The authors also show how such a $CO_2$ reduction process occurs in the experiment and how this characteristic might be used to restrict concurrent hydrogen evolution. Furthermore, researchers demonstrate that the catalyst's overpotential and associated turnover frequency (TOF) for $CO_2$ reduction match favorably with the best molecular porphyrin-based catalyst in the literature [137]. The authors proposed the mechanism of their study in Figure 11a. As a result, the authors believe that these findings could have significant implications for the development of new and improved molecular catalyst electrodes, as well as the formulation of optimized process conditions for efficient electrochemical $CO_2$ reduction to CO and the reduction of products to a greater degree. Cobalt porphyrins were investigated in order to take advantage of the

frameworks' distinctive features, such as charge carrier mobility due to π–π stacking and the stability afforded by covalent bonding and reticular geometries [140]. When deposited on porous carbon fabric for electrochemical testing, the COFs had a catalytic onset potential of −0.42 V vs. RHE in $CO_2$ saturated carbonate electrolyte and a maximal activity of −0.67 V vs. RHE, generating CO predominantly at a FE of 90%. With hydrogen evolution being favored at increasingly negative potentials, this strong selectivity for the $CO_2$RR begins to decrease.

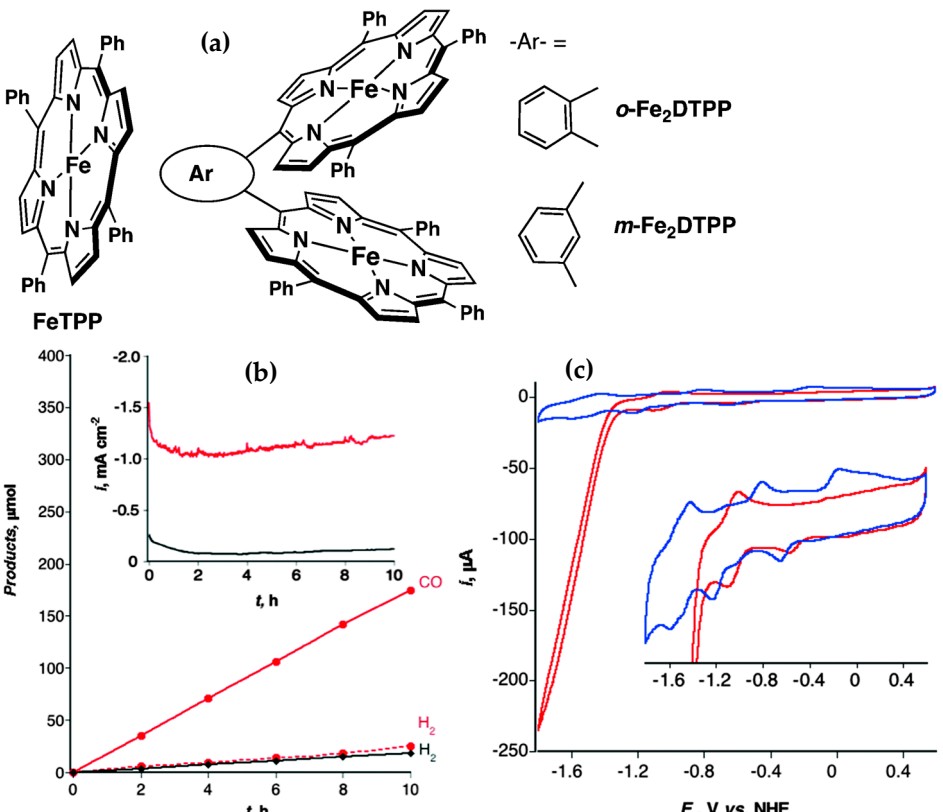

**Figure 10.** (**a**) Chemical structures of the iron porphyrin monomer, FeTPP, and iron porphyrin dimers, o−$Fe_2$DTPP and m-$Fe_2$DTPP. (**b**) CVs of o−$Fe_2$DTPP at 100 mV scan rate in DMF/10% $H_2O$ containing 0.1 M TBAPF$_6$ supporting electrolyte under Ar or $CO_2$. Inset: magnified trace of CVs. (**c**) $CO_2$ reduction products with time and the current density–time profile (inset) produced during the 10 h chronoamperometry experiment at −1.35 V vs. NHE in a DMF/10% $H_2O$/0.1 M TBAPF$_6$ solution saturated with $CO_2$ without (black lines) and with 0.5 mM o−$Fe_2$DTPP (red lines). (Reprinted with permission from Ref. [128]. Copyright 2015 Royal Society of Chemistry).

COFs were synthesized with longer linkage molecules to increase the spacing between porphyrins for greater pore volume, as illustrated in Figure 1b, in order to improve their performance. As a result, more electrolyte access and $CO_2$ adsorption were possible, resulting in a lower onset potential (−0.40 V) and a 2.2-fold increase in catalytic activity at −0.67 V. For graphene-enhanced $CO_2$ diffusion restrictions, as well as Re-porphyrins, were discovered to have a considerable impact on activity [144]. In comparison to non-stirring circumstances, stirring to boost $CO_2$ diffusion into the electrolyte increased the reaction rate by three times while maintaining selectivity. This is inextricably tied to the fact that, whereas active centers are clearly defined in these materials, active site density is often low, necessitating effective use of active sites.

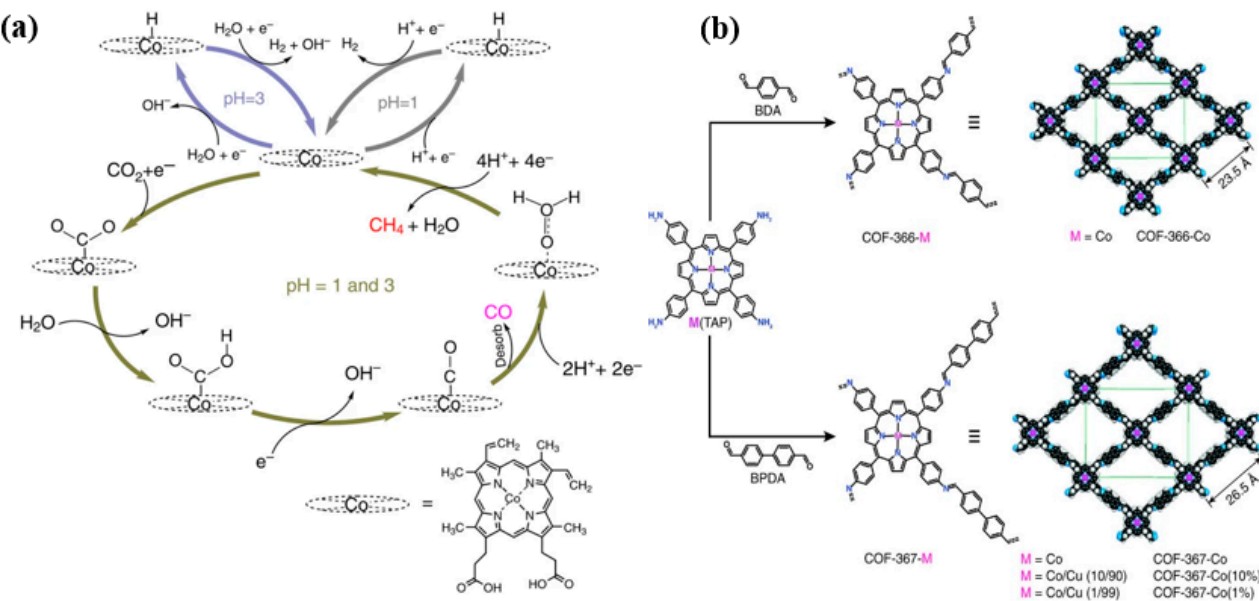

**Figure 11.** (**a**) Proposed mechanistic scheme for the electrochemical reduction of $CO_2$ on Co protoporphyrin. (Reprinted with permission from ref. [126]. 2015, American Association for the Advancement of Science). (**b**) Design and synthesis of metalloporphyrin-derived 2D covalent organic frameworks. Materials Studio 7.0 was used to generate space-filling structural models of COF-366-M and COF-367-M, which were then improved with experimental PXRD data. (Reprinted with permission from Ref. [126]. Copyright 2015 American Association for the Advancement of Science).

Wang and coworkers [145] have used the synthesis of silver MOF-mediated as a simple and scalable method to prepare highly dispersed supported Ag catalysts with very low metal loadings for $CO_2$ER in GDE configuration. The resulting Ag-coordination polymer was directly grown in carbon paper with a microporous layer. The remarkable activity of these catalysts reached a Faradaic efficiency of more than 96% for CO production at an outstanding current density of 300 mA cm$^{-2}$. This technology of using MOF materials as mediators to prepare ordered metallic and doped structures have known considerable attention in the past decade [146–150].

### 2.5. Phthalocyanines-Based Catalysts

Lu and coworkers [151] have used cobalt phthalocyanines (CoPC) as an electrocatalyst for $CO_2$ reduction to CO. The authors have used microflow cell configurations at low voltages using a KOH catholyte. The cell was operating at 0.26 overpotential, reaching nearly 94% Faradaic efficiency. The partial current density of $CO_2$-TO-CO reached 31 mA/cm$^{-2}$. This study outperforms the previously reported cobalt phthalocyanine in the H-Cell configuration, which yielded only $-6$ mA/cm$^{-2}$ total current density and a Faradaic efficiency of 93% of CO [152]. Previously reported literature has suggested that incorporation of CoPc in a coordination polymer such as poly-4-vinylpyridine (P4VP) can lead to a selective $CO_2$ reduction activity over hydrogen evolution reaction, which is a $CO_2$RR competing reaction [142,153–155].

More recently, Liu et al. [156] have revealed the effect of using the P4VP coordination polymer encapsulation on $CO_2$RR activity. Indeed, authors have shown that the rate-determining step in the $CO_2$ mechanism is axial coordination from the pyridyl moieties in poly-4-vinylpyridine to CoPc while the HER was suppressed due to the sluggish and weak proton transport through the polymer. Whereas earlier, another team used a scanning tunnel microscope to unveil the $CO_2$RR mechanism over CoPC catalysts, the authors have revealed that making the Co$^I$PC-$CO_2$ intermediate is the rate-limiting step [157]. X. Zhang [158] has elaborated on carbon materials supported by CoPC for $CO_2$ reduction; the CoPC/CNT material had the best Faradaic efficiency of CO production (90%) at a

current density of $-10$ mA/cm$^{-2}$ at $-0.63$V vs. RHE. The carbon black/CoPc and reduced graphene oxide/CoPc showed less than a third of the current density at $-0.59$ V vs. RHE and a lower FE(CO) as well as low reusability. The authors have yet to improve their performance of the CoPC/CNTs by using the Cyano-substituted CoPc hybrid approach and the prepared catalyst was noted as CoPc-CN/CNT. This substituted catalyst had a current density of $-15$ mA/cm$^{-2}$ and FE(CO) of 95%. The CNT hybridization acts as a conducting platform, allowing the easiness of electron transfer and enabling the high degree of catalytic site exposure that leads to high current densities. The cyano groups facilitate the electron withdrawal to form the Co$^I$ responsible for $CO_2$-to-CO reduction.

Copper phthalocyanines (CuPC) have been studied by several groups for their interesting properties for $CO_2$ reduction properties. Latiff and coworkers [159] have developed a long (10–30 μm) and thin (10–30 nm Ø) carbon nanotubes supported by CuPC structures that can efficiently reduce $CO_2$ with better Faradaic efficiencies. The CuPC/CNTs yielded an overall Faradaic efficiency of 66.3% for $C_1$ and $C_2$ byproducts, with CO being the highest evolving product with a Faradaic efficiency of nearly 44% (Figure 12).

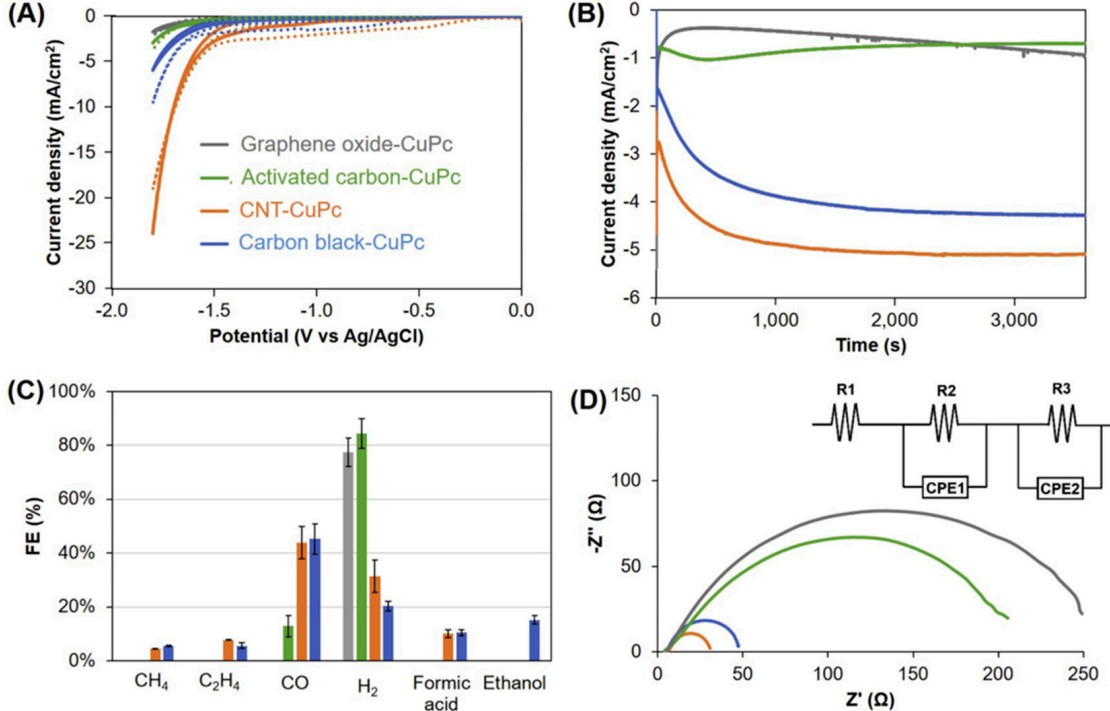

**Figure 12.** (**A**) CV measurements of various carbon-supported copper(II) phthalocyanines in $CO_2$-saturated (marked in continuous lines) and $N_2$-saturated (marked in dotted lines) 0.5 M KCl solution. (**B**) Chronoamperometry measurements of the materials over 1 h under an applied potential of $-1.05$ V vs. RHE. (**C**) Corresponding product analysis results from chronoamperometry runs. (**D**) EIS spectra for the materials under study are shown by their Nyquist plots, where CPE stands for constant phase element (Reprinted with permission from Ref. [159]. Copyright 2020 Elsevier).

B. Mei et al. have used operando elucidation of the dynamic and structural transformation of CuPc during $CO_2$RR electrolysis (Figure 13). Authors have applied operando high-energy resolution fluorescence-detected X-ray absorption spectroscopy to reveal the responsible phenomenon for their observed $C_2H_4$ formation. They have suggested that the main phenomenon taking place is the aggregation of Cu sites (making copper clusters) with applied potential, leading to the byproduct's formation [160]. Whereas S. Kusama and coworkers reached high electrochemical $CO_2$ reduction to $C_2H_4$ with a Faradaic efficiency of 25% by using bare crystalline CuPC supported by a conductive carbon black at $-1.6$ V vs. Ag/AgCl. The CO and $CH_4$ were also produced but with low Faradaic efficiency when a limited electrolysis time was performed; however, when long-term 12 h electrolysis

was performed, the CO evolution became more pronounceable, terminating the $C_2H_4$ evolution [161].

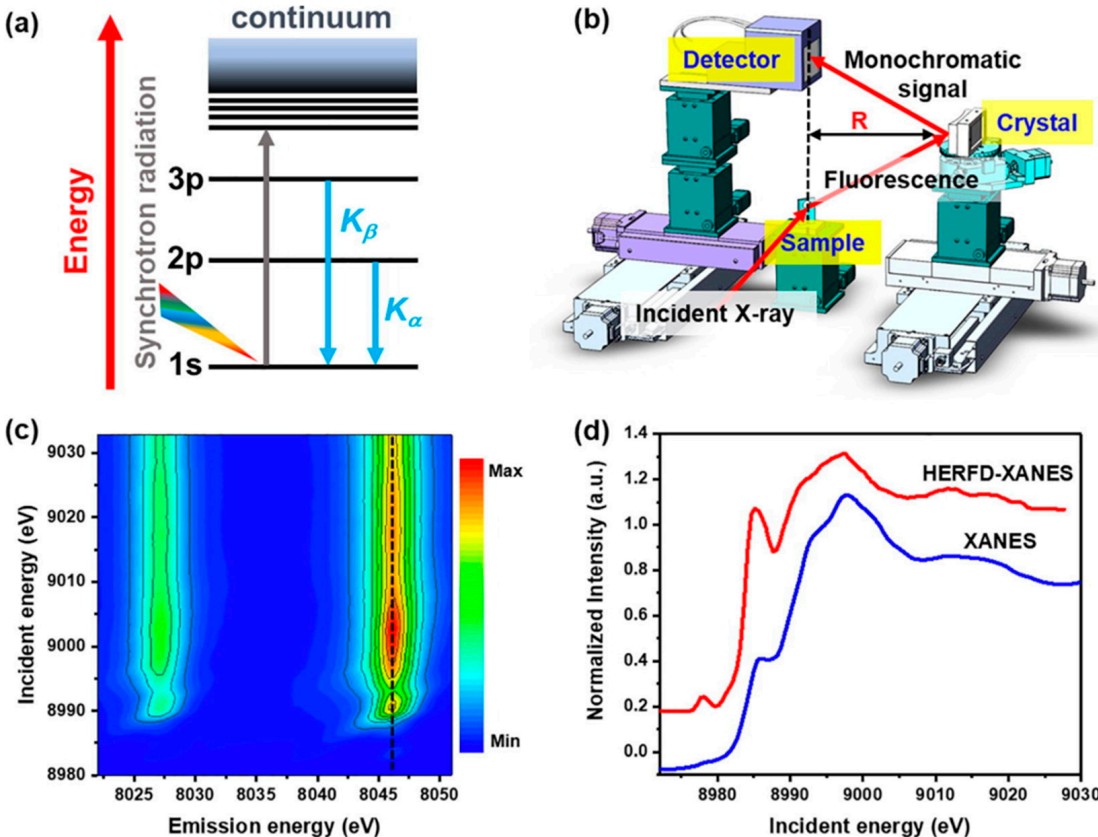

**Figure 13.** (**a**) Energy scheme of $K\alpha$ and $K\beta$ emission lines. (**b**) Experimental setup; (**c**) $K\alpha$-RIXS plane of CuO around the Cu K-edge. The contour planes at the emission energy of ~8026 and 8046 eV are the $K\alpha_2$ and $K\alpha_1$-RIXS plane, respectively. The black-dashed line located at constant emission energy of 8046.3 eV. (**d**) HERFD-XANES (red) and conventional XANES (blue) spectra of CuO (Reprinted with permission from Ref. [160]. Copyright 2022 Elsevier).

Other metal phthalocyanines have also been studied for $CO_2$ reduction electrolysis, such as tin phthalocyanine [162], nickel phthalocyanines [163], iron phthalocyanines [164], and zinc phthalocyanines [165]. In the case of tin phthalocyanine dichloride, the resulting aggregation hybrid catalyst can catalyze the electroreduction of $CO_2$ to HCOOH and CO at a total Faradaic efficiency of ca. 90%. The active site of tin phthalocyanine dichloride responsible for electrocatalytic $CO_2$ reduction is confirmed to be metallic Sn, whose local structure is strongly affected by the adjacent macrocyclic ligands. The dispersed NiPc molecules unlocked remarkable electrocatalytic properties for the $CO_2$RR, unlike the aggregated form. The molecular dispersed electrocatalyst NiPc–OMe exhibited FE(CO)s of >99.5% over a wide current density range of $-10$ to $-300$ mA cm$^{-2}$ and stable performance at the practically relevant current density of $-150$ mA cm$^{-2}$ for 40 h [163].

The iron phthalocyanine electrocatalytic performance of FePc-G, FePc-Gr, FePc-R, FePc-R/$H_2O_2$, FePc/G heterostructures, FePc nanorods, and graphene were investigated to elucidate the role of iron valence degree +II and +III. The optimal catalyst exhibited a high FECO of >90% at about $-0.5$ V vs. RHE and an onset potential of $-190$ mV, and syngas production is easy to obtain and depends mainly on the potential. Furthermore, the DFT simulation revealed that for $CO_2$RR, the catalytic activity of Fe(II)Pc should be better than Fe(III)Pc, and that of Fe(II)Pc/Fe(III)Pc dimer higher than individual Fe(II)Pc or Fe(III)Pc [164]. Finally, ZnPc/carbon nitride nanosheet hybrid catalysts could be effectively excited by visible light for PEC-$CO_2$RR. The major product was methanol, and the

highest methanol generation efficiency was achieved at −1.0 V. The methanol yield was 13 μmol. L$^{-1}$ after 8 h. The mechanisms for the three different $CO_2$RR processes, including PC, EC, and PEC-$CO_2$RR, are proposed. In PEC-$CO_2$RR, ZnPc/-carbon nitride nanosheets exhibited a synergic effect and methanol production efficiency is much better PC- and EC-$CO_2$RR [165].

*2.6. $CO_2$ Reduction Mechanisms*

Depending on the three classes of electrocatalysts that were highlighted in the sub-Section 2.1. Different $CO_2$ value-added products can be obtained. The target byproducts are controlled by tuning the binding energy of the adsorbed intermediate such as (CO*, COOH*, CHO*, COH*) (Figure 14). For example, when the interaction between the surface of the electrocatalysts and the reduction intermediates is not strong enough, CO and formate are the main reduction products (Type 1 and 2). Inversely, when the electrocatalysts bind strongly with CO*, this will be further reduced to other by-products (Type 3) [166]. This is where copper shows an incredible ability to transform $CO_2$ into hydrocarbons, The basic explanation for its ability to generate products other than CO is that Cu binds *CO neither too weakly nor too strongly [37].

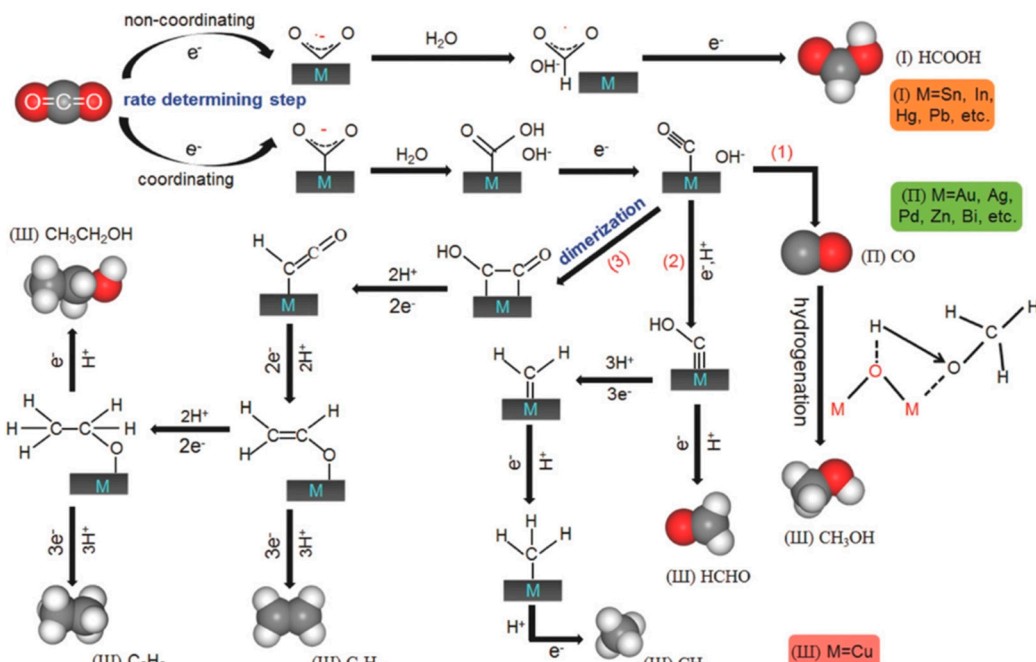

**Figure 14.** Schematic mechanism of different metal electrocatalysts for $CO_2$ reduction reaction in aqueous solution. (Reprinted with permission from Ref. [165]. Copyright 2019 Elsevier).

**3. Conclusions and Prospects**

The Electrochemical $CO_2$ reduction technique is still in its early stages compared to other $CO_2$ conversion technologies, although it is being actively researched. For electrochemical $CO_2$ reduction, a range of materials have been investigated as catalysts, and the catalysts should be modified depending on the intended products. Nanostructured catalysts should be further adjusted in light of the cell design of gas-diffusion electrodes. To date, the majority of investigations have employed pure $CO_2$ that has been concentrated; however, for practical applications, the conversion of dilute $CO_2$, especially in the presence of possible catalyst poisons such as S compounds, should be researched more intensively. In addition, several technological hurdles remain in $CO_2$ electrochemical reduction technology, including (i) inadequate catalyst activity, (ii) limited product selectivity, and (iii) insufficient stability. Our technology appears to be far from adequate when it comes to the actual application of $CO_2$ reduction to creating useable low-carbon fuels. Low catalyst stability

appears to be the current key barrier to industrial-scale deployment. As a result, the main emphasis of effort in this field remains the development of highly active, selective, and stable electrocatalysts for $CO_2$ reduction.

**Author Contributions:** H.A.A.: Conceptualization; Writing the draft, Data curation; Formal analysis; Project administration. M.Z.: Writing the draft, review & editing; Data curation; Formal analysis. A.B.: Writing—review & editing. M.A.: Writing—review & editing. All authors have read and agreed to the published version of the manuscript.

**Funding:** This research received no external funding.

**Acknowledgments:** This work was supported by the faculty of sciences and the Mohammed V University in Rabat.

**Conflicts of Interest:** The authors declare no conflict of interest.

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
