# Peer review of "CO2 Electroreduction over Metallic Oxide, Carbon-Based, and Molecular Catalysts: A Mini-Review of the Current Advances"

_catalysts, doi:10.3390/catal12050450_

Round 1

Reviewer 1 Report

Ahsaine and coworkers reported a review of CO2RR electrocatalysts, discussing the enhancement of CO2RR electrocatalytic activity by developing metal simple substance, metal alloy, metal oxide, metal single-atom and metal-organic compound catalysts. In this paper, the research progress of CO2RR catalysts is introduced to a high extent. However, there are ambiguities and deficiencies in the article that need to be improved and corrected. Before the final publication, the authors should consider the following comments:

1. In the section of Introduction, the third paragraph seems illogical. This article is an overview of electrocatalytic CO2RR, and it is abrupt and illogical for the author to mention the advantages and disadvantages of photocatalysis here.

2. It is known that there are several catalytic mechanisms for CO2RR and that the products vary in different electrolyte environments. It is suggested that the authors add a description of the relevant catalytic mechanisms.

3. What are the advantages of CO2RR for metal oxides compared to metal simple substances and alloys?

4. In subsection 2.1, the authors cite a large number of Au- and Cu-based alloys and simple substance catalysts. It is strange for the authors to write such a large section on the topic. What is the connection between this chapter and the subject of this article?

5. A mechanistic description of the selectivity about single-atom catalysts when involved in CO2RR catalysis is missing. Moreover, it is incomplete for the authors only present Cu single-atom catalysts in this chapter.

6. As a review of cutting-edge research in the field, the authors cite literatures from five years ago in bulk. Is this in line with the "current advances" in the title?

Author Response

We thank the reviewer 1 for his pertinent comments and suggestions which helped us to improve the paper. Please find below our responses to your queries. The changes are marked in red color in the manuscript.

Replies to Reviewer 1

Ahsaine and coworkers reported a review of CO2RR electrocatalysts, discussing the enhancement of CO2RR electrocatalytic activity by developing metal simple substance, metal alloy, metal oxide, metal single-atom and metal-organic compound catalysts. In this paper, the research progress of CO2RR catalysts is introduced to a high extent. However, there are ambiguities and deficiencies in the article that need to be improved and corrected. Before the final publication, the authors should consider the following comments:

  1. In the section of Introduction, the third paragraph seems illogical. This article is an overview of electrocatalytic CO2RR, and it is abrupt and illogical for the author to mention the advantages and disadvantages of photocatalysis here.

Answer: Thank you for this important comment. We have deleted the part of photocatalysis in the third paragraph.

  1. It is known that there are several catalytic mechanisms for CO2RR and that the products vary in different electrolyte environments. It is suggested that the authors add a description of the relevant catalytic mechanisms.

Answer: Thank you. We have added a paragraph discussing the CO2RR mechanisms.

  1. What are the advantages of CO2RR for metal oxides compared to metal simple substances and alloys?

Answer: This is a very important question and it is still under debate in the electrocatalysis community especially in the CO2 electroconversion. Both metals and metal oxide-based catalysts have been reported decades ago for the CO2RR reaction because they are easy, relatively stable and cheap to synthesize and they do not require special preparation conditions compared to molecular, MOFs and COFs electrocatalysts. The difference between these two classes of catalysts resides in their chemical engineering aspect to achieve high energy efficiency and selectivity. This is being said, serious preparation control of metal or metal oxide structures in morphological tailoring, atoming scale structuration, guided alloying, electrolyte design, crystallites interconnection, and strained surfaces is needed for better electrocatalytic activity. Another challenging task is that in some cases notorious to mild reconstruction in the reaction conditions, which makes it hard to describe the catalytic cycles processes without operando elucidation.

  1. In subsection 2.1, the authors cite a large number of Au- and Cu-based alloys and simple substance catalysts. It is strange for the authors to write such a large section on the topic. What is the connection between this chapter and the subject of this article?

Answer: We have also mentioned and discussed metal and bimetallic electrocatalysts. However, knowing that copper is the only known metal capable of producing different range of hydrocarbons or a mixture of chemicals that are of a big interest of industry, we have put much interest in copper metals and oxide in this paragraph.  Again, we should note that we have added some paragraphs on other metal/bimetallic catalysts and new references. The scope is limited here. If we are to discuss all the metal/bimetallic and oxide electrocatalysts, it will be another full review. In this section (subsection 2.1,) we are focusing on the catalysts that are presenting a big challenges and activity.

  1. A mechanistic description of the selectivity about single-atom catalysts when involved in CO2RR catalysis is missing. Moreover, it is incomplete for the authors only present Cu single-atom catalysts in this chapter.

Answer: Thank you for this comment. We have added other SACs for CO2 reduction such as Ni, Iron and Co.

  1. As a review of cutting-edge research in the field, the authors cite literatures from five years ago in bulk. Is this in line with the "current advances" in the title?

Answer: This has been addressed according to both reviewers 1 and 2. Recent references have been added.

We thank the editor and the reviewer for his constructive comments and we hope that our revised version is to your satisfaction and that our manuscript can be accepted for publication in Catalysts.

Reviewer 2 Report

This mini-review is about CO2 reduction covering metal oxide, other material-based and molecular catalysts. The topic is of high interest and therefore the review would attract readers. However, the overall quality of the work should be improved before publication in Catalysts, the current form contains inattentive errors that do not meet the standards of this journal.

  1. There are too many typos and inconsistent writing modes in the text. These should be eliminated. For example, the use of capital letters in the name of elements, wrong sub- and suprscripts, mixed use of different abbreviations, wrong use of the genitive form, bold letter for names and figures in the text, etc.
  2. The introduction promises highlights on recent progress in the field, but in spite of that less than 10% of the references comes from the past 3 years. Please, add some more recent studies, too. Some important literature is missing, for example C. Janaky's excellent achievements.
  3. In some cases the negative sign is missing from the potential values.
  4. All figures are reprinted from others' works. This is OK, but the captions in Figs. 3, 4, 6-9, are way too short to explain the contents for the readers.
  5. A few tables would be definitely needed to summarize the data listed in the text.

Author Response

We thank the reviewer 2 for his pertinent comments and suggestions which helped us to improve the paper. Please find below our responses to your queries. The changes are marked in red color in the manuscript.

Replies to Reviewer 2

Comments and Suggestions for Authors

This mini-review is about CO2 reduction covering metal oxide, other material-based and molecular catalysts. The topic is of high interest and therefore the review would attract readers. However, the overall quality of the work should be improved before publication in Catalysts, the current form contains inattentive errors that do not meet the standards of this journal.

  1. There are too many typos and inconsistent writing modes in the text. These should be eliminated. For example, the use of capital letters in the name of elements, wrong sub- and suprscripts, mixed use of different abbreviations, wrong use of the genitive form, bold letter for names and figures in the text, etc.

Answer: Thank you. All these remarks have been checked and corrected carefully in the manuscript.

  1. The introduction promises highlights on recent progress in the field, but in spite of that less than 10% of the references comes from the past 3 years. Please, add some more recent studies, too. Some important literature is missing, for example C. Janaky's excellent achievements.

Answer: Thank you for this important comment. We are aware of the works of Janaky and we have added some valuable references of the Janaky’s group.

  1. In some cases, the negative sign is missing from the potential values.

Answer: Thank you. This error has been corrected.

  1. All figures are reprinted from others' works. This is OK, but the captions in Figs. 3, 4, 6-9, are way too short to explain the contents for the readers.

Answer: Thank you. We have added the description of each figure as requested.

  1. A few tables would be definitely needed to summarize the data listed in the text.

Answer: Thank you. We are aware that tables are a media of summarizing to the readers. However, we aimed for a mini-review on the advances and most performing class of electrocatalysts on each section and if we inserted tables on each section; it will lengthen our review. If the reviewer and editor insist to put tables, we will be happy to provide long tables in each section.

We are preparing a thorough review on SACs in this regard.

We thank the editor and the reviewer for his constructive comments and we hope that our revised version is to your satisfaction and that our manuscript can be accepted for publication in Catalysts.

Round 2

Reviewer 1 Report

The paper has been revised according to reviewers' comments. It can be accepted now.

Reviewer 2 Report

The authors made most of the requested changes. I do not insist on the tabulated listing of catalysts performance data. The manuscript can be published in the present form.